# The ISJ 3D Brace, a Providence Brace Evolution, as a Surgery Prevention Method in Idiopathic Scoliosis

**DOI:** 10.3390/jcm10173915

**Published:** 2021-08-30

**Authors:** Luis González Vicente, María Jiménez Barrios, Josefa González-Santos, Mirian Santamaría-Peláez, Raúl Soto-Cámara, Juan Mielgo-Ayuso, Diego Fernández-Lázaro, Jerónimo J. González-Bernal

**Affiliations:** 1Orthopedic Department, Institut Sant Joan SL, 08009 Barcelona, Spain; luisgonzalez555@hotmail.com; 2Department of Health Sciences, University of Burgos, 09001 Burgos, Spain; mspelaez@ubu.es (M.S.-P.); rscamara@ubu.es (R.S.-C.); jfmielgo@ubu.es (J.M.-A.); jejavier@ubu.es (J.J.G.-B.); 3Department of Cellular Biology, Histology and Pharmacology, Faculty of Health Sciences, Campus of Soria, University of Valladolid, 42003 Soria, Spain; diego.fernandez.lazaro@uva.es; 4Neurobiology Research Group, Faculty of Medicine, University of Valladolid, 47002 Valladolid, Spain

**Keywords:** idiopathic scoliosis, nighttime orthotic treatment, surgery, quality of life

## Abstract

Background: The high incidence of idiopathic scoliosis worldwide as well as the serious health problems it can cause in adulthood, make it necessary to seek effective treatments to prevent the progression of the disease to more aggressive treatments such as surgery and improve patients’ quality of life. The use of night braces, besides a less severe influence on the patient’s quality of life, is effective in stopping the progression of the curve in idiopathic scoliosis. Methods: A longitudinal study was performed with an experimental population of 108 participants who attended orthotic treatment at the University Hospital of Barcelona, with ages between 4 and 15 years old, with a main curvature greater than 25 degrees and a Risser between 0 and 3. The participants received treatment with Providence ISJ-3D night braces until their pubertal change (mean duration of 2.78 years for males and 1.97 years for females). Results: The implementation of night-time orthotic treatment in children with idiopathic scoliosis is effective in slowing the progression of the curve and in the prevention of more aggressive treatments such as surgery, maintaining the patient’s quality of life. Conclusions: The use of night braces is efficacious in the treatment of idiopathic scoliosis, although new studies including more sociodemographic data as well as curves from 20 degrees of progression are necessary.

## 1. Introduction

Idiopathic scoliosis, with a worldwide incidence ranging from 0.47% to 5.2% [1], is the most common type of scoliosis in children and adolescents during their growth phase, without an apparent cause. It is defined as any deformity of the spine characterized by a lateral deviation greater than 10 degrees (Cobb angle) [2] combined with a spinal rotation and generally associated with a hypociphosis. Depending on the age of onset, it can be classified as infantile up to two years old, juvenile, between three and nine years old, and adolescent, from ten years old or higher [3,4,5].

Factors such as the patient’s chronological and bone age, Risser’s sign, and menarche have been linked to the patient’s growth, and thus to the progression of scoliosis [6]. The Risser sign determines the skeletal maturity of the child, providing an estimate of how much skeletal growth remains, due to it classifies the progress of bone fusion of the iliac process. Risser degrees range from 0 to 5 where: 0 means no ossification, degree 1 up to 25% ossification, degree 2 between 26% and 50% ossification, degree 3 between 51% and 75% ossification, degree 4 between 76% and 100% ossification and degree 5 is related to the complete bone fusion of the process [6,7,8,9]. The severity of scoliosis is assessed by the Cobb angle or degrees of spinal curvature [7], so that if this angle exceeds the “critical threshold” (30°–50° at the end of the growth stage), there is a greater risk of health problems, decreased quality of life, aesthetic deformity, disability, pain and functional limitations in adulthood [6].

One of the most effective conservative techniques in the treatment of scoliosis is the use of a brace, which improves the patient’s balance and stabilizes the curvature, preventing its progression, especially in the higher risk phases: Risser’s sign of 0–1 or pre-menarchal phase in the female, and Risser’s sign 3 in the male [10,11,12,13].

On the other hand, surgery is indicated in those cases in which the child or adolescent presents a structured primary curve with a Cobb angle greater than 45°, that is to say, unacceptable curves in an adult due to the effect they could have of different physiological functions [7,14].

With regard to orthopedic treatment, there are different types of braces depending on aspects such as the material they are made of, the time of use or the body region they cover. Regarding the time of use, a distinction is made between daytime braces (20–23 h) and nighttime braces (8–10 h), which are similarly effective in the treatment of scoliosis, the only difference being that nighttime braces have less impact on the quality of life and less psychosocial impact, which in practice translates into greater adherence to and better compliance with treatment [15,16,17,18,19,20,21,22,23].

The Providence brace is a night-time scoliosis correction system, designed and manufactured through a computer-assisted CAD-CAM system, which makes it easy to use and increases patient comfort [16,24]. The ISJ 3D brace, designed through the Rodin computer system, is an evolution of the Providence brace, which like that one, is based on the correction of deformities through compression forces at three or four points, improving the result of the treatment by avoiding the appearance of secondary curves. The brace is made through the CAD-CAM system starting from the lumbar point or 0 point, located between the iliac crest and the twelfth rib, applying a lateral-rotational pressure on the patient’s spine. This system allows obtaining a personalized module for each patient, with a precise virtual correction for its subsequent manufacture [25].

The operation of this brace is based on the premise that scoliosis has a biomechanical mode of deformity progression based on the Hueterr Volkamann principle, whereby greater axial compression slows growth of the deformity and less axial compression accelerates growth [26].

The double curve design of the night brace produces a controlled application of opposite, direct, lateral and rotational forces on the trunk that leads to a push of the apexes of the curves towards the midline or more. In addition to these pressure points, it also has empty or hollow areas that allow to mantained the shell in continuity. The carbon fiber reinforced braces included in this type of corset offer greater patient comfort because the empty areas space is reduced. On the other hand, this three-point system also helps to control double curves allowing the treatment of curves with apex at the height of T6 without the need for a neck extension [27].

Depending on the location of the curve, derotation occurs differently. The lumbar pad that the corset presents between the iliac crest and the twelfth rib performs the segmental derotation of the curve at the lumbar level [27]. Derotation of the thoracic part is facilitated by the position that the patient adopts in the supine or prone position on the bed, since his own weight is transferred in the form of pressure to the costovertebral joints producing a change in the curvature of the spine [28]. Derotation of this section of the device is achieved thanks to the CAD/CAM model whereby at first the thoracic section is separated from the lumbar section and then the thoracic portion is rotated a certain amount to realign with the lumbar section [27].

Given that there is no effective method for the detection of scoliosis progression and that the use of full time braces has been shown to have variable adherence as a consequence of the psychosocial wear and tear they produce, the present study has been designed with the objective of verifying the effectiveness of the ISJ 3D Night Brace, a Providence evolution, in the treatment of idiopathic scoliosis as a method of preventing surgery.

## 2. Materials and Methods

### 2.1. Design and Study Sample

A non-controlled clinical trial was designed with a single intervention group (*n* = 108), whose study population was made up of all patients of both sexes, aged between 4 and 15 years, who during the development of the study attended the Orthopedic Surgery and Traumatology Outpatient Service of the University Hospital (Barcelona, Spain), and who met the following inclusion criteria: they had been diagnosed with idiopathic scoliosis, had a main curvature greater than 25° (the upper limit of curvature included in the study is 47 degrees (*n* = 1), and three other users had 44 degrees), with a Risser’s sign of 0 to 3, and had not previously received any surgical treatment. Cases of neuromuscular or congenital scoliosis (they usually do not tolerate the use of the corset and require other types of orthopedic measures), those presenting a Risser’s sign of 4–5 or those who had already completed the bone growth stage were excluded.

### 2.2. Procedure

All of volunteers were invited to participate in the study, having to sign the informed consent if they accepted, thus respecting the confidentiality and anonymity of each participant, in accordance with the Organic Law 15/1999 on the protection of personal data, being assigned to each of them an ethical identification code of the San Joan de Deu Foundation.

After the general physical analysis, the spine was examined and the characteristics of the deformity were recorded, using a plummet to measure the imbalance of the trunk in relation to the pelvis [29].

A radiographic evaluation (one at the beginning, one with the corset in place, and then every 6 months, up to one year after the removal of the corset) was carried out using the Cobb method [30]. According to the International Scientific Society for Orthopedic and Rehabilitation Treatment of Scoliosis (SOSORT) a curve progression of less than 5° is considered a successful treatment [31]. Based on the topography elaborated by means of the structured light technique, the Posterior Trunk Symmetry Index (POTSI) and the Deformity in the Horizontal Plane Index (DHOPI) were determined [32].

After the initial assessment, CAD-CAM measurements were taken using the latest generation of digital scanners.

The evaluation was carried out at three timepoints: a first evaluation before starting the treatment, a second evaluation at the end of the treatment and a final evaluation one year after its completion. This evaluation consisted of the measurement according to the Cobb method of the different types of curves (PT, MT, TL/L), identification of the apex of each main curve, classification of the different curves according to Lenke, as well as the measurement of the Risser degree. In addition, at the end of the treatment, the CAVIDRA quality of life scale was administered to people with spinal deformity, in order to check how the use of the corset has influenced the patients’ quality of life.

### 2.3. Outcome Measures

Through the results obtained from primary curvature as the main variable of the study, the participants who would require surgery were determined. The magnitude of this curve was established by Cobb angle measurements carried out by a group of specialists who followed a previously established protocol.

The magnitude of the curve was determined by measuring the Cobb angle, derived from an X-ray of the standing spine. The Cobb angle is the angle formed by a line drawn perpendicular to the top of the upper vertebrae of the scoliotic curve and a similar perpendicular line drawn along the bottom of the lower vertebrae [33]. The result obtained is a numerical value that allows both the comparison of the magnitude of each curve, and the comparison between each of them. A curve progression of 6 or more degrees would be considered treatment failure [31,34].

To carry out the evaluation, on the one hand, Lenke’s classification was determined considering the pattern of the curve, specifying the limits and name of the scoliosis according to the segment involved, six curve types are distinguished in this system based on whether the Proximal Thoracic (PT), Main Thoracic (MT), and Thoracolumbar/Lumbar (TL/L) regions are major, minor structural, or nonstructural including: Type 1, MT; Type 2, Double Thoracic (DT); Type 3, Double Major (DM); Type 4, Triple Major (TM); Type 5, TL/L; Type 6, TL/L-MT.; On the other hand, taking into account the convexity of the main curve, it was possible to distinguish whether the curve was right or left, and finally, the degree of bone growth (Risser) was established [35].

### 2.4. Statistical Analysis

The total sample consisted of 108 patients (94 females and 14 males). The mean age of the participants at the beginning of the treatment was 10.14 for the boys (DS ± 2.77), and 11.17 for the girls (DS ± 2.28), 72.22% of the total were between 10 and 13 years old. To verify the relationship between the different variables analyzed, the data was analyzed using the Pearson Correlation. The means obtained at the different time points were compared through the Wilcoxon and Manova test. Statistical analysis was carried out with the IBM SPSS Statistics, Version 25.0 (IBM Corp, Armonk, NY, USA) using a value of 0.05 as the limit of statistical significance.

## 3. Results

First, data were obtained from a total of 326 cases, but after a purging process, those whose questionnaires were incomplete, that is, they lacked data in some of the evaluations, were discarded in order to obtain as faithful and clean a sample as possible.

A total of 108 participants received treatment with the ISJ-3D Night Brace, which they had to wear between 8 and 10 h a night (M = 9.25 h; SD ± 2.18) every day until reaching Risser 4 in the girls and Risser 5 in the boys, which would establish the end of the treatment. The average age at treatment completion was 13.73 years for boys (SD ± 1.19) and 13.74 years for girls (SD ± 1.21) except for one patient who completed treatment at 11 years because she underwent the change prematurely at age 9. The data at the beginning of the treatment were: mean height 144.96 (SD ±12.99), mean weight 41.69 (SD ±7.83), mean BMI 19.62 (SD ±1.45). The Cobb angle of the main curve was M = 2932 SD ± 539 at pretest, and M = 3200, SD = 1307 at the end of treatment. The duration of treatment with the brace has been 2.78 years for boys (SD ± 1.85) and 1.97 years for girls (SD ± 1.19). The follow-up time was one year after the end of the treatment. The adherence of the brace was mesured by an Orthotimer, an electronic wearing time system that thoroughly documents the wearing time of the device. Each brace has a built-in a small sensor with a unique individual ID number and can be automatically connected to the patients registered in the database. This sensor is controlled with a wireless reading device which is connected with the computer via a USB plug and the saved wearing time dates are transferred to the software. The wear time is presented in various charts. A total wear time overview or an individually selected range extracted from the entire monitoring period of time can be observed [18,24,36].

The descriptive statistics for topographic and radiographic variables of initial assessment 1 and final assessment 2 are summarized in Table 1.

The values obtained in the different types of curves (PT, MT, TL/L) before and after treatment with the night brace are summarized in Table 2.

When the pre-test values were compared with the post-test values of the degrees of the different types of curves at the beginning and end of the treatment, statistically significant differences were observed in the PT (Z = 3.42; *p* = 0.001), MT (Z = 3.47; *p* = 0.001) and TL/L (Z = 2.26; *p* = 0.024) curves, in such a way that the degrees of all the curves have increased, although in no case reaching 6 degrees. When checking the percentage of users who improved, meaning that the degrees at the end of the treatment are lower than at the beginning of the treatment, the proximal dorsal curves are the ones that improved the most with this orthopedic treatment, obtaining an improvement in 40% of the cases (43 of 108 patients). The MT curves were the ones that improved the least after the treatment, with a greater improvement observed in 29% of the cases (31 of 108 patients). With respect to the lumbar curves after finishing the treatment, 33% (38 of 108 patients) of the participants improved.

All the curves (PT, MT, TL/L) evolved less than 6 degrees (1.59 < $\bar{x}$ < 5.53), with no significant differences according to the type of curve (*p* = 0.95), that is, the orthotic treatment is equally effective on the PT, MT and TL/L curves. In the same way, the results indicated statistically significant changes (*p* = 0.000) in the evaluation carried out one year after finishing the treatment in relation to the evaluation made at the beginning of the treatment, in such a way that the main curve increased until reaching 36.11 degrees of average (SD ± 14.33) against 29.72 (SD ± 5.45) obtained at the end of the treatment. With respect to the overall curve, 28% showed an improvement of 5°. On the other hand, the ANCOVA according to the type of Lenke indicated that there were no significant differences, between the post-test and the pre-test (F_(5, 101)_ = 0.34, *p* = 0.883). It is observed that Lenke type 5 that reduces the degrees, and Lenke type 1, are the types that best respond to treatment (Table 3).

The descriptive results showed that the MT curves increased in Risser 0 and 1 and decreased in Risser 2 and 3, in the same way as occurred with the TL/L curves (Table 4).

13% of the participants who received the night brace treatment required surgery. Of all of them, 50% correspond to juvenile scoliosis and the other 50% to adolescent scoliosis, with the average age of treatment initiation being 10–50 years. Of the 14 patients who needed surgery, 11 were women and three were men. It is important to highlight the similarity between the 14 surgical curves, since 13 of them were double curves, right dorsal, left lumbar and one of them right dorsal-lumbar. In all of them, the predominant surgical curve was the right dorsal thoracic curve (Figure 1).

With respect to the secondary measures, the Quality of life in Spinal Deformities (CAVIDRA) or Quality of Life Profile for Spinal Deformities (QLPSD), developed in Spain by Climent in 1995, was used to measure the quality of life in patients with spinal deformities at growth age. It is a self-administered questionnaire consisting of a total of 21 items distributed in five dimensions: psychosocial functioning, sleep disorders, back pain, body image and spinal mobility. The patient shows his or her agreement or disagreement degree on a five-grade on a Likert type scale, being able to obtain a total score between 21 and 105, where a higher score means a greater impact on the quality of life. The reliability obtained from the scale for this study was 0.910 [37,38].

The results of the evaluation of the quality of life according to the Cavidra scale showed that the incidence of the ISJ-3D corset evolution of the Providence brace on the patients’ quality of life is very low, since a mean score of 1.42 over 5 has been obtained (SD ± 0.42). In the same way, the domains of the scale in an unidividualized way presented little repercussion on the quality of life, being the highest incidence in the domain of body image with a mean of 1.81 (SD ± 0.84) and the lowest in the psychosocial functioning with a mean of 1.23 (SD ± 0.27). On the other hand, no statistically significant differences (*p* = 0.00) were observed in the results of the Cavidra in relation to the degrees of the main curve measured in the three temporalis moments (beginning, end and at the year of treatment). In the results obtained, a significant correlation was observed between the degrees of the main curve and the scores of the different dimensions and the totals of the Cavidra scale (*p* = 0.001), in such a way that higher grades are related to higher scores in the scale, which means a worse quality of life (Table 5).

Table 6 presents the quality of life scores according to the Lenke classification, it is observed that the lowest scores correspond to type 1 (1.20, SD = 0.20) and type 5 (M = 1.24, SD = 0.45), which they are the groups that improve the most. The differences are significant, F_(5, 102)_ = 5.28, *p* = 0.001.

## 4. Discussion

The choice of treatment for idiopathic soliosis is mainly based on the size and location of the deformity. Different studies have shown the effectiveness of orthopaedic appliances in stopping curve progression and correcting it [39,40]. The prescription of the brace treatment is directed to patients with a curve between 24° and 40° still growing, curves less than 25° and progression from 5° to 10° in six months or patients with scoliosis between 20° and 25° with skeletal immaturity (Risser 0) [41].

In the study carried out in the United States and Canada, the effectiveness of the Boston-type thoracolumbosacral orthosis in the progression of idiopathic scoliosis was verified. A total of 242 patients participated in the trial, of which 60% received braces and 40% were only observed. The brace was prescribed for 18 h per day. The treatment failure rate was 25% with an average time of brace use of 12.1 ± 6.5 h per day. The study concluded that this type of device reduces the progression of the curves and a greater number of hours of corset use is associated with greater benefits [42].

According to these results, in other studies carried out to verify the effectiveness of Wilmington and Boston braces with a sample of 36 patients with a mean age of 12 years and a sample of 395 with a mean age of 13.2, respectively, concluded that the progression of the curve was reduced only in those participants who use the brace for more than 20 h per day [43,44].

In the same way, in the study conducted by Sanders et al., where the use of the brace was prescribed between 17 and 23 h per day in 126 patients with adolescent idiopathic scoliosis, the results obtained showed that this treatment can avoid surgery in patients provided that the treatment regimen is followed. However, only 31% of the participants wore the brace for 10 h or more and 17% for more than 14 h. Among the factors that can influence compliance with treatment, the discomfort of wearing it, the social problems it causes, and poor self-image stand out, with patients choosing to assume the risk of progression to surgery rather than wear an orthopedic appliance for that number of hours. Thus the weak evidence of the effectiveness of bracing can be explained in part by poor compliance with the brace use [44,45].

With respect to the efficacy of nighttime orthotic treatments, the study by Lee et al., verified the efficacy of the Charleston night brace for the treatment of idiopathic scoliosis in 95 patients aged 10–16 years with curves between 25 and 40 degrees and concluded that 84.2% of patients had 5 degrees or less progression of the curve, 15.8% had a progression of 6 or more degrees and only 7.8% progressed to surgery [46]. According to these results in the study carried out by Simony et al., which verified the effectiveness of treatment with the Providence night brace in curves between 20 and 42 degrees, concluded that it was effective in preventing the progression of the curve in 71 of 80 patients and only five participants were referred to surgery, making these results comparable to the full-time treatment with the Boston corset [24].

In a study by Yrjönen et al., the effectiveness of the full-time Boston-style corset and the Providence night-time corset was compared in a total of 72 patients. The average correction of prone positioning braces was 92% for the Providence night brace and 50% for the foot X-ray group and only one of the patients treated with the Providence device progressed to surgery, which is in agreement with the results obtained in the present study [16].

With respect to the psychosocial study of the present study, the results showed that the use of the night brace was effective in the treatment of the progression of the curve without influencing the patient’s quality of life, unlike other full time orthopedic devices with which the same results are achieved in the progression of the curve only if the time of use prescribed is complied with, having a great influence on the patient’s quality of life [11,16,36,47].

The results obtained in this research with the use of the ISJ-3D evolution of Providence night brace coincide with those obtained by other authors in which they demonstrate the same results as full-time Boston-type orthoses, but unlike these, night braces do not influence the patient’s quality of life, which makes treatment adherence greater. In addition, due to the rotational component of the ISJ-3D night brace, it is able to achieve better scoliosis corrections than other night braces such as the Charleston that do not have this component [18,24,39,43,48,49].

The results obtained in the study by Bonilla et al., showed that the different evaluation instruments designed to evaluate the quality of life of patients with idiopathic scoliosis, such as the SRS-22r and CAVIDRA, present adequate psychometric properties with regard to the domain of perceived self-image [50]. However, according to the studies carried out by Rezaei et al. or Liu et al. other assessment instruments such as the Bad Sobernheim stress questionnaire (BSSQ) or the Braces Questionnaire (BrQ), are more specific for measuring the quality of life of patients with brace treatment in adolescent idiopathic scoliosis, present adequate reliability and validation psychometric properties and are excellent tools to measure this aspect [51,52].

The study’s findings should be considered within the context of its strengths and limitations. Among its strengths is the collection of data from a homogeneous sample in terms of age, Risser degree and degrees of progression of the curve or its longitudinal design following up one year after the completion of treatment. With regard to the limitations of the study, the absence of a control group for observation or treatment with orthopedic apparatus on a full-time basis that would allow the results to be compared stands out, as well as the lack of data collection such as the body mass index, the socio-demographic state or the vertebral rotation. In addition, the CAVIDRA assessment was used to evaluate the quality of life because of its high reliability and its validation into Spanish however, it has not been specifically evaluated, validated or analyzed for this pathology. Nevertheless, this tool is not specific for patients who receiving conservative orthopedic treatment such as BRQ or BSSQ, which will be taken into account for future studies.

## 5. Conclusions

The fundamental idea of night braces is to reduce the negative effects of full brace use on the quality of life without worsening the end results of treatment. The results obtained with the ISJ-3D night brace, an evolution of the Providence night brace, showed the great effectiveness of this type of orthosis since only 13% of the participants required surgery and the average increase in the curves in no case exceeded 6 degrees, without affecting the patient’s quality of life. The results also showed the same efficacy of the treatment regardless of which curve was predominant.

Taking into account these results and the scarce number of studies evaluating the efficacy of this type of orthosis, the development of new research on this subject is justified, with representative samples, with a Risser 0 and in curves between 20 and 40 degrees.

## Figures and Tables

**Figure 1 jcm-10-03915-f001:**
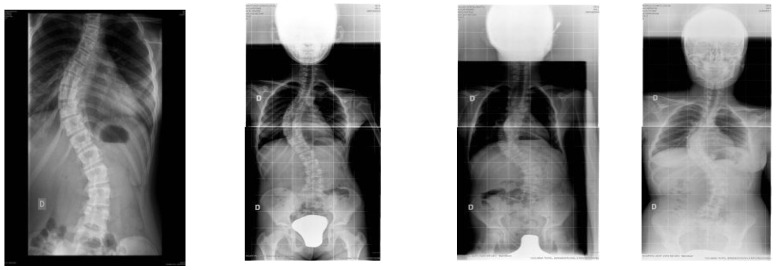
X-ray of the 14 participants who required surgery.

**Table 1 jcm-10-03915-t001:** Descriptive statistics (initial assessment 1-final assessment 2) for topographic and radiographic variables.

	Evaluation	M	SD
ATR thoracic hump	1	5.22	3.51
	2	5.67	4.22
ATR lumbar hump	1	4.65	2.57
	2	5.01	3.55
Main cobb	1	29.32	5.39
	2	32.00	13.07
Vertebral rotation	1	8.53	5.22
	2	8.90	5.36
Thoracic kyphosis angle	1	27.48	7.31
	2	27.66	7.66
Lumbar lordosis angle	1	45.21	7.75
	2	46.89	8.61
DIHP	1	6.22	1.33
	2	6.86	1.61
PTSI	1	20.01	8.63
	2	20.91	8.88
CP	1	40.33	8.05
	2	41.05	8.17

ART = angle of rotation of the trunk; Cobb princ. = Cobb angle of the main curve of scoliosis; Rot. apex = angle of rotation of the apical vertebra of the main curve of scoliosis. DIHP = deformity index in the horizontal plane; CP = columnar profile; PTSI = posterior trunk symmetry index.

**Table 2 jcm-10-03915-t002:** Values and comparison of the degrees of each type of curve, before and after performing brace intervention (Mean = 2.07 years, SD = 1.3).

	Pre-Intervention		Post-Intervention		Student’s T	*p*-Value
	M	SD	M	SD		
**(PT)**	12.12	8.13	15.44	10.60	3.58	0.001
**(MT)**	27.26	7.19	32.48	15.56	3.84	0.000
**(TL/L)**	21.85	8.26	25.36	12.66	2.73	0.008

PT = Proximal Thoracic; MT = Main Thoracic; TL = Thoracolumbar; L = Lumbar; M: mean; SD: standard deviation; *p*-value < 0.05.

**Table 3 jcm-10-03915-t003:** Ancova test, differential score between the post-test and the pre-test (2-1), Lenke clasification.

LENKE Type	Mean	Std. Deviation	N
1	1.34	6.99	29
2	2.90	326	11
3	6.28	15.65	21
4	5.69	8.66	23
5	−0.25	6.48	12
6	5.58	14.39	12
Total	3.68	10.32	108

*p* = 0.883.

**Table 4 jcm-10-03915-t004:** Differential score between the post-test and the pre-test (2-1), on the main curve.

Risser	Mean	Std. Deviation	N
0	4.40	11.25	82
1	2.86	6.25	15
2	0.00	6.52	8
3	−2.00	4.35	3
Total	3.68	10.32	108

**Table 5 jcm-10-03915-t005:** Values of Pearson’s correlation between degrees of main curve and Cavidra scale values.

	Main Curve		
	Pearson’s Correlation	*p*-Value	N
Cavidra total	0.95	0.001	108
Psychosocial functioning	0.78	0.001	108
Sleep disorder	0.87	0.001	108
Back pain	0.74	0.001	108
Body image	0.88	0.001	108
Mobility	0.83	0.001	108

*p*-value < 0.05.

**Table 6 jcm-10-03915-t006:** Anova, Outcome measures related to the CAVIDRA score (QLPSD) according to the type of curve in the Lenke classification.

Lenke	N	Mean	SD
1	29	1.20	0.20
2	11	1.36	0.34
3	21	1.47	0.44
4	23	1.61	0.45
5	12	1.24	0.45
6	12	1.73	0.44
Total	108	1.42	0.42

*p* = 0.001.

## Data Availability

The data that support the findings of this study are available from the corresponding author upon reasonable request.

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
