# Peer review of "The ISJ 3D Brace, a Providence Brace Evolution, as a Surgery Prevention Method in Idiopathic Scoliosis"

_jcm, 2021, doi:10.3390/jcm10173915_

Round 1

Reviewer 1 Report

The present study reports the effectiveness of the ISJ 3D night brace as a method of prevention of surgery in idiopathic scoliosis. The authors propose its use for 8-10 hours during every night until reaching Risser 4 in girls and Risser 5 in boys. It is an interesting paper which adds more information for conservative treatment of idiopathic scoliosis. There are a few issues in the methods and the results which need to be clarified and therefore some revisions are necessary prior to publication.

  1. In the introduction section, line 40 the word “new” should be replaced with the appropriate age.
  2. In Materials and Methods, paragraph 2.1, line 84 it is reported that the patients with curvature greater than 30 degrees were included. What is the upper limit of the curvature?
  3. In Materials and Methods, paragraph 2.1, lines 85-87 the authors included in the study children with Risser 0-2 and excluded children with Risser 4-5. There is no information for patients with Risser 3.
  4. In Materials and Methods, paragraph 2.2, lines 128-130 the authors used the CAVIDRA quality of life scale to check if the use of the corset has influenced the patients’ quality of life after completing the treatment. It would be more interesting to measure quality of life during treatment when children were wearing the ISJ 3D night brace because the purpose of the study is to show the effectiveness of the brace itself and not the overall impact of the spinal deformity on patients’ quality of life.
  5. In Materials and Methods, paragraph 2. lines 152-161 the authors used the CAVIDRA questionnaire which is a tool to “measure the quality of life in patients with spinal deformities at growth age”.  Does the CAVIDRA questionnaire measure quality of life in relation to brace treatment? There are other disease specific tools in the literature which are more appropriate for this purpose and it would be more interesting if the authors had used one of these tools.
  6. In the results section, line 174 the authors report that “those whose questionnaires were incomplete were discarded…”. What are these questionnaires?
  7. In the results section, line 189 and Table A there are three curve types. In contrary in the Material and Methods (lines 143-151) the authors carried out the evaluation using Lenke’s classification which has four curve types. Which of the above is correct?
  8. In the results section, lines 193-201 the authors analyze the effect of brace treatment in curve progression and found that “the dorsal curves are the ones that have improved the most”. What do the authors mean by the term “improved”?
  9. In the results section, lines 208-209 the authors report that “the descriptive results showed that the distal curves increased in Risser 0 and 1 and decreased in Risser 2 and 3…”. Please provide these descriptive results.
  10. The authors must provide in the Results section all the relevant information which they describe in the Materials and Methods section. There are no data regarding clinical evaluation, inclinometer and topographic measurements of the patients as well as Cobb angle measurements.
  11. In the Discussion section, lines 293-294 the authors report “that lack of data collection such as the body mass index, the socio-demographic state or the vertebral rotation are limitations of the study”. From the Materials and Methods section it is clear that most of that information is available. Addition of that information would add strength to the present study.

Author Response

First of all, we would like to express our sincere gratitude for all comments and suggestions received from the Reviewer 2. This information has certainly enriched the text for its best understanding, thank you very much indeed. We have clarified the reviewer’s questions. We have introduced the required changes both in our answers to the specific comments and in the final manuscript v2.

Reviewer 2 Report

Dear Sir or Madame,

in the manuscript "ISJ 3D brace evolution of Providence, as a method of prevention of surgery in idiopathic scoliosis" the authors present their bracing method for the treatment of idiopathic scoliosis in adolescents. The treatment method is not new, but always interesting to review, since whole day cast or corset treatment is a great impairment for the patients.

The article is of interest to orthopedic surgeons and pediatricians. However, the study lags a control group and the follow-up period does not become clear to me. Altogether, I recommend to accept the manuscript after major revision.

For the revision process, the following paragraphs need amendment:

  • Ll. 20: "besides not influencing the patient´s quality…" – a brace is still a great impairment. I recommend to write "less severe" etc.
  • Ll. 23: delete the institution´s name from the manuscript, this needs to be done throughout the whole manuscript

  • Material & Methods: The section needs to be shortened. Only the essentials to understand the study and its design belong in this section:
  • N=108
  • Follow-up period?, how many follow-up x-rays were performed and in which period of time?
  • Ll. 108 "According to the SOSORT .. improvement of 5 degrees or more." – a curve progression of less than 5° is considered a successful treatment? Or what does the authors mean?
  • Ll. 143: Please classify the curves in an international accepted manner – Lenke classification is recommended
  • LL152: the paragraph belongs to the results.
  • Ll. 160/61 delete the sentence.
  • Ll. 163 – replace women and men with male / female or boys and girls (-> to results)
  • Paragraph 163-171 can significantly be shortened to the essential information
  •  
  • Results
  • This paragraph needs more structure to clearly indicate the study´s results to the reader n=108, follow up time, mean cobb angle, correction / halt of progression by cast, patient-related outcome measurements (CAVIDRA and QLPSD Score)
  • Treatment time = follow-up ?
  • What happened to patients, who had failure of treatment – did they receive surgery or was the cast/corset treatment extended to 23 hours a day?
  • Table A: please mention the wear time of the braces
  • The questionnaires were answered by the patients themselves or by their parents?

  • Discussion
  • 234-236: delete
  • Why were patients with neurological scoliosis excluded?
  • Why was the treatment on-set for boys and girls almost the same 13.73 vs. 13.74 years. – Usually, girls reach puberty faster and treatment for scoliosis, incl. surgery, at a younger age than boys.
  • How many surgeries were prevented due to wearing of the brace in the observed cohort?

Author Response

(The authors gave the same response as above.)

Round 2

Reviewer 2 Report

the authors did a good job and improved the manuscript to my full satisfaction

Author Response

Thank you very much for your appreciation. We are very satisfied with the reviewers' recommendations, and we believe that the manuscript has greatly improved thanks to their input.
It has been a pleasure to be able to review the article with such precise and accurate considerations.

Thanks a lot.